# Role of IFN-γ from Different Immune Cells in *Chlamydia* Infection

**DOI:** 10.3390/microorganisms13102374

**Published:** 2025-10-15

**Authors:** Xuan Chen, Wenjing Yang, Yuchen Hu, Yang Zhou, Zhou Zhou

**Affiliations:** Hunan Provincial Key Laboratory for Special Pathogens Prevention and Control, Hunan Province Cooperative Innovation Center for Molecular Target New Drug Study, Institute of Pathogenic Biology, Hengyang Medical School, University of South China, Hengyang 421001, China; cxadventure@163.com (X.C.); yangwenjing52023@163.com (W.Y.); 13135002023@163.com (Y.H.); abc84480750@163.com (Y.Z.)

**Keywords:** *Chlamydia*, infection, IFN-γ, innate immunity, adaptive immunity

## Abstract

*Chlamydia* invades multiple mucosal tissues in humans and animals. The body’s first line of defense against chlamydial infection is provided by innate immunity, whereas adaptive immunity plays a crucial role in managing the infection’s progression and preparing the immune system to combat reinfection. Host resistance to chlamydial infection necessitates a coordinated effort between innate and adaptive immune cells. Numerous cell types are capable of producing interferon gamma (IFN-γ) as a defense mechanism against chlamydial infection, thereby effectively mediating the clearance of infection. However, the distinct roles of various immune cell populations in responding to chlamydial infection, as well as their functions during infection progression, remain poorly understood. Therefore, we will discuss the various roles of IFN-γ released by different immune cells in chlamydial infection, focusing primarily on experimental animal models and a review of available data from in vivo cellular immunological studies in humans.

## 1. Introduction

*Chlamydia* refers to a group of Gram-negative microbes characterized by a unique developmental cycle, allowing them to penetrate bacterial filters [1]. *Chlamydia* is a bacterium that must live inside host cells and possesses own its enzymatic systems; however, it cannot generate the energy necessary for its metabolic processes [2]. They not only rely on triphosphates and intermediate metabolites of host cells as energy sources, but also require tryptophan, phospholipids, nucleotides, and other metabolites to survive [3]. This suggests that *Chlamydia trachomatis* has evolved to minimize its genome size at the expense of complete independence, making it highly dependent on its host. This parasitic relationship often leads to disease in humans.

*Chlamydia* exhibits a distinctive developmental cycle composed of two stages, switching between the infectious elementary body (EB) and the replicative reticulate body (RB) [4]. These two forms are adapted for intracellular replication and extracellular survival, respectively. Despite having a similar genome and common Immunobiological characteristics, different types of *Chlamydia* cause different diseases [2,5]. For example, the primary source of infectious blindness globally is *C. trachomatis*, which is responsible for trachomatous conjunctivitis [6]. Additionally, *Chlamydia* in the genital tract can lead to various diseases, including ectopic pregnancy, pelvic inflammatory disease, and tubal edema [7].

*Chlamydia* infection often recurs or persists, indicating the lack of sterilizing immunity [8]. Innate immunity is the first line of defense in host immunity, responding quickly to the pathogens invading the host, which plays a significant role in the process of host resistance to chlamydial infection. Adaptive immunity includes cellular and humoral immunity, such as the cloning and amplification of antigen-specific T cells and B cells. Chlamydial infection of epithelial cells triggers complex immune cell interactions. The infected cells respond by rapidly secreting cytokines like interleukin (IL)-1, IL-6, interferon-gamma (IFN-γ), and tumor necrosis factor-alpha (TNF-α) upon bacterial activation. These cytokines facilitate the movement of immune cells, including macrophages and neutrophils, toward the site of infection by stimulating the release of chemokines (CXCL1/2, CCL2/5, CXCL9/10) and enhancing the expression of adhesion molecules (ICAM-1, VCAM-1). IFN-γ specifically enhanced Th1 and NK cell recruitment by stimulating CXCL9-11 production [9,10]. These substances activate or attract immune cells to initiate or enhance inflammation in response to *Chlamydia* [11,12]. The immune system utilizes these cytokines to curb *Chlamydia* proliferation and manage the infection, ultimately aiding in the prevention or reduction of the progression of *Chlamydia*-related disease [13,14,15].

Many studies have demonstrated that IFN-γ is the most effective cytokine against chlamydial infection [16,17]. The mechanisms by which *Chlamydia* induces IFN-γ and the subsequent impact of IFN-γ induction on chlamydial infection and pathogenicity are intriguing areas of investigation. Notably, the endogenous components of *Chlamydia* may act as potential activators for IFN synthesis. *Chlamydia* DNA can be recognized by the cytoplasmic DNA receptor CGAS and induce type I interferon production through the STING-TBK1-IRF3 pathway. Meanwhile, *Chlamydia* LPS is recognized by TLR2/4, and heat shock protein 60 (CHSP60) can be recognized by TLR4, promoting the expression of interferons and inflammatory factors [18,19,20].

It is worth noting that the pro-inflammatory cytokine IFN-γ regulates the immune response to chlamydial CHSP60. In vitro, the expression of CHSP60 was increased in the presence of a low concentration of IFN-γ. However, during persistent *C. trachomatis* infection, the immune response to CHSP60 is enhanced, suggesting that these mechanisms could be significant in the process of infection in vivo [21]. In addition, IFN-γ affects the course of infection by down-regulating the expression of c-Myc, a key regulator of host cell metabolism. This regulation is dependent on the STAT1 signaling pathway. Notably, restoration of c-Myc expression was able to reverse the IFN-γ-induced state of persistent chlamydial infection in both cell lines and human fallopian tube organoids, confirming the importance of this regulatory pathway during infection [22].

Several studies on *Chlamydia* infection have shown that many cells can produce IFN-γ, including natural killer T (NKT) cells, Th1 (CD4) or Th1-like (such as CD8) cells, NK (killer ILC1s), ILC1s (helper ILC1s), Ex-ILC3s, mucosal-associated invariant T (MAIT) cells, and even some myeloid cells [23,24,25]. However, during chlamydial infection in different mucosal tissues, the specific cell types responsible for delivering IFN-γ close to *Chlamydia*-infected cells remain unknown. Here, we offer a summary of the role of IFN-γ, which is generated by different cell types in response to *Chlamydia* infection, along with the related pathogenesis [24,26]. Furthermore, we discuss the various roles of IFN-γ released by different immune cells in chlamydial infection, focusing primarily on experimental animal models and a review of the available data generated from in vivo cellular immunological studies in humans.

## 2. The Role of Immune Cells in Host Resistance Against Chlamydial Infection

When the host is invaded by *Chlamydia*, host cells are capable of initiating both innate and adaptive immune responses to combat *Chlamydia* infection by recognizing the antigens released by *Chlamydia* during various phases of its developmental cycle. T cells, MAIT cells, NKT cells, and innate lymphoid cells (ILCs) are critical components of the immune response against *Chlamydia* infection.

### 2.1. MAIT Cells

Recent research on the mechanism of host anti-infection has revealed the crucial role of interactions between innate and adaptive immunity in the control and elimination of infectious diseases. The initial barrier against *Chlamydia* infection is innate immunity, which typically activates during the early phase of the infection. MAIT cells are involved in managing microbial infections, which may depend on their ability to eliminate infected cells in response to various microbial challenges.

MAIT cells are conserved and abundant αβ-TCR T lymphocytes, which are primarily found in the lamina propria of the intestinal mucosa and exhibit Th1-like cytotoxicity against various microbial infections [25]. At the same time, under the stimulation of bacteria and fungi, MAIT cells are capable of generating multiple cytokines, including IFN-γ and TNF-α, which promote the antimicrobial response (Figure 1) [27,28,29].

MAIT cells differ from traditional T cells in that they express a restricted T-cell receptor (TCR) lineage, characterized by an unchanged T-cell receptor chain [30,31]. While traditional T cells typically react to polypeptide antigens presented by MHC molecules, MAIT cells specifically recognize microbial vitamin derivatives that conventional T cells cannot detect. Vitamin B2, also known as riboflavin, along with its metabolites, plays a crucial role in energy production across various organisms. Nevertheless, due to the inability of mammals to produce vitamin B, the distinctive capability of MAIT cells to recognize metabolites derived from the vitamin B biosynthesis pathways in bacteria and yeast suggests that riboflavin synthesis serves as a marker for microbial activity that activates MAIT cells. Riboflavin serves as a precursor for the flavin coenzymes FMN and FAD, which are crucial for cellular REDOX metabolism [32]. *Chlamydia*, as an obligate intracellular bacterium, is highly dependent on the host for its metabolism and may not be able to synthesize riboflavin itself. However, riboflavin can be obtained through the host transport mechanism to meet their own metabolic needs [2]. Based on this, we suggested that riboflavin metabolism of *Chlamydia* might be associated with the activation of MAIT cells, but further verification is needed.

Le Bourhis and colleagues demonstrated that a range of bacteria and fungi, such as *Candida albicans*, *Pseudomonas aeruginosa*, *Candida glabrata*, *Lactobacillus acidophilus*, *Saccharomyces cerevisiae*, *Klebsiella pneumoniae*, and *Staphylococcus aureus*, are capable of activating MAIT cells [27]. Moreover, MAIT cells have the ability to indirectly sense microbes through the recognition of Microbe-Associated Molecular Patterns (MAMPs). When APCs are activated by either native or synthetic ligands interacting with TLR2, TLR3, TLR4, TLR5, TLR8, and TLR9, they release pro-inflammatory cytokines that can stimulate MAIT cells to secrete IFN-γ [33,34,35]. Simultaneously, pro-inflammatory cytokines have the ability to either spontaneously activate intrinsic T cells or function as a co-signaling mechanism to amplify the weak signals relayed by the TCR [36].

Wang et al. characterized the plasticity of MAIT cell subsets in vivo, with distinct IFN-γ-producing MAIT1 and IL-17A-producing MAIT17 subsets [37]. After undergoing adoptive transfer or during acute infections with *Legionella* or *Francisella*, RORγt^+^ MAIT 17 cells can transition into RORγt^+^ T-bet^+^ MAIT 1/17 cells and RORγt^−^T-bet^+^ MAIT 1 cells. This transformation is essential for controlling intracellular pathogens, as it allows MAIT 17 cells to acquire the capability to produce IFN-γ.

Numerous studies employing MR1-deficient mouse models have demonstrated the critical role of MAIT cells in controlling bacterial infections. The absence of this lymphocyte population renders the host significantly more susceptible to pathogens such as *Klebsiella pneumoniae* [38]. Moreover, upon exposure to infectious agents, MAIT cells can swiftly generate large amounts of pro-inflammatory cytokines, thereby regulating the effector functions of both innate and adaptive immune cells [39]. During the early phase of infection, MAIT cells are promptly activated through recognition of microbial riboflavin-derived metabolites, exerting innate-like effector responses. However, their functional contribution is not limited to early immunity; in later stages of infection, they continue to accumulate at sites of infection and profoundly shape adaptive immune responses through sustained cytokine secretion and modulation of Th1-type immunity [40]. Furthermore, MAIT cell function is markedly influenced by their tissue tropism and local microbial environment: not only pathogens but also commensal-derived ligands can modulate their activity, indicating a dual role for these cells in both host defense against pathogens and maintenance of microbial homeostasis [41].

In conclusion, as a group of T cell subsets that have evolved conservatively, MAIT cells are capable of responding to a majority of bacteria by recognizing metabolites from the vitamin B2 biosynthesis pathway via TCR-mediated interactions, subsequently activating ligands of MAIT cells. These compounds are exclusive to bacteria and yeasts that produce riboflavin, enabling MAIT cells to exhibit extensive antimicrobial activity. Experimental models of various microbial infections demonstrate the protective role of MAIT cells, further confirming their antimicrobial capabilities.

MR1 is a highly conserved monomorphic molecule related to class I of the major histocompatibility complex that is widespread in mammals [42]. Research indicates that gut microbiota is critical for the development of MAIT cells; these cells are lacking in germ-free mice, but colonization with riboflavin-producing *E. coli* can restore their development and expansion [43]. In addition, MAIT cells are not only dependent on gut microbiota for proper function, but also require interaction with MR1-expressing B cells. The absence of MAIT cells in patients who lack B cells reinforces the vital role of MR1 in the selection and expansion of MAIT cells [44,45]. These observations highlight the significant roles that MAIT cells play within the immune system, with IFN-γ being a crucial factor in that context.

However, there are few studies on whether *Chlamydia* can activate MAIT cells and enhance T-cell response. Research conducted on humans indicates that MAIT cells are found more often in the intestinal and jejunal mucosa, comprising 5% of CD4 T cells, as well as in the liver parenchyma and the vascular system (accounting for 20% of T cells). Epithelial cells are the main target cells of *Chlamydia* infection, and MAIT cells play a key role in antimicrobial immunity. In mycobacterial infections, MAIT cells can be activated through the MR1-dependent pathway [46]. Viruses lack the riboflavin synthesis pathway [47]. Their APCs can produce pro-inflammatory cytokines such as IL-12 and IL-18, thereby stimulating or activating MAIT cells in response to infections. Notably, intestinal macrophages are susceptible to chlamydial infection, as evidenced by the detection of *Chlamydia* in macrophages and enteroendocrine cells from patients with severe irritable bowel syndrome (IBS) [48]. Considering the enrichment characteristics of MAIT cells in mucosal tissues and their immune function against intracellular pathogens, we speculate that these cells could be crucial in orchestrating the immune response to *Chlamydia* infection, but this mechanism has not yet been confirmed.

### 2.2. NKT Cells

Natural killer T (NKT) and natural killer (NK) cells are essential elements of the innate immune system, playing significant roles in various diseases while also linking innate and adaptive immunity. NKT cells represent a distinctive subset of lymphocytes that can function as various immune cells, including B cells, T cells, dendritic cells, NK cells, and macrophages, and can also serve as effector cells to kill tumor cells. NKT cells possess both T-cell surface markers and characteristic markers typical of NK cells, such as NK1.1 (NKR-P1A) and Ly49 family molecules that inhibit killing [49].

Rather than surveying peptide–MHC complexes, NKT cells operate under CD1d restriction and lock onto glycolipid cues displayed by CD1d itself. CD1d molecules present glycolipid antigens—exemplified by α-galactosylceramide (α-GalCer)—to the invariant T-cell receptor of NKT cells, establishing a highly specific recognition event [50]. This glycolipid–CD1d ligation triggers robust activation of NKT cells within only two to three hours, a response markedly faster than that of conventional T cells [51]. Once activated, NKT cells rapidly unleash a synchronized burst of IFN-γ and IL-4 within minutes to hours, generating an early cytokine storm [52]. The rapid and abundant IFN-γ/IL-4 milieu subsequently directs the differentiation of naïve CD4^+^ T cells toward either a Th1 or Th2 phenotype, thereby promoting the functional execution of downstream adaptive immunity [53].

Although NKT cells constitute < 1% of peripheral αβ T cells, within 2–6 h post-infection, they rapidly detect microbial glycolipids or virus-elicited endogenous ligands via CD1d-restricted recognition. In vitro, α-GalCer or bacterial-derived glycolipids such as α-glucuronosylceramide potently activate human and murine CD1d-restricted invariant NKT (iNKT) cells, driving robust proliferation and copious IFN-γ secretion. In vivo studies corroborate that within 24 h post-infection, early iNKT-cell activation accompanied by marked IFN-γ and IL-4 release is readily detectable. Moreover, utilizing NKT-deficient models—namely CD1d^−/−^ and Jα18^−/−^ mice—the study demonstrates the obligate contribution of this lymphocyte subset to early pathogen clearance: pulmonary bacterial burdens in Sphingomonas capsulata-infected knockout animals increase 12-14-fold at early time points, with significantly delayed bacterial elimination [54]. This evidence shows that, although the NKT cell population is very small, it plays a powerful role in the immune response.

First, in a respiratory infection model with *Chlamydia pneumoniae*, Jα18-knockout mice, which lack iNKT cells, exhibit markedly greater weight loss, more severe pulmonary pathology, and significantly higher bacterial loads in the lungs. However, the same deficiency strategy yields opposite results in an intra-articular *C. trachomatis* infection: CD1d-knockout mice instead display milder joint pathology and lower bacterial burdens [55,56]. Most strikingly, the *Chlamydia muridarum* model shows that CD1d- or Jα18-knockout mice develop attenuated pulmonary inflammation and reduced bacterial loads; conversely, deliberate activation of iNKT cells with α-GalCer exacerbates pulmonary infection and induces oviduct fibrosis [57,58]. Mechanistically, iNKT cells are rapidly activated and, via IFN-γ secretion and CD40L–CD40 interactions, license dendritic cells—especially the CD8α subset—to produce abundant IL-12 [59]. This cytokine milieu polarizes naïve T cells toward IFN-γ–producing Th1/Tc1 effector cells, thereby establishing a robust type-1 protective immune response that clears intracellular *Chlamydia*. Collectively, the functional outcome of NKT cells in chlamydial diseases is determined by the pathogen species-specific antigenic repertoire, host genetic background, and the resultant Th1/Th2/IL-10 polarization axis. For example, activated NKT cells have opposite effects on *C*. *pneumoniae* infection and *C*. *muridarum* infection. The former is inhibited, and the latter is aggravated. Analysis of iNKT cells indicated strikingly different key cytokine patterns, with a response dominated by IFN-γ to *C. pneumoniae*, while the response to *C. muridarum* was primarily driven by IL-4 [60]. Furthermore, compelling evidence indicates that iNKT cells bridge innate and adaptive immunity by modulating dendritic cells (DCs) function. In vitro and in vivo studies consistently demonstrate that DCs isolated from Jα18-knockout mice—lacking iNKT cells—fail to confer protective immunity upon adoptive transfer into recipient mice. In contrast, when iNKT cells are present, CD8α^+^ DCs become functionally “licensed,” resulting in markedly increased IL-12 production that potently drives naïve T-cell differentiation into IFN-γ–producing Th1/Tc1 effector cells [61,62].

These studies showed that NKT cells contribute to the immune response in a manner that resembles the function of regulatory T cells, thus driving downstream immune responses, activating inflammatory responses, or inducing immune tolerance. Notably, Armitage et al. further confirmed that wild-type female mice exhibited higher *Chlamydia* burden, more severe tubal obstruction, and elevated inflammatory cytokine levels than CD1d^−^/^−^ (NKT-deficient) mice during urogenital *Chlamydia* infection. In contrast, Jα18^−^/^−^ (iNKT-deficient) mice had normal infection progression. This suggests that non-invariant NKT cells activated by CD1d-presenting lipid antigens mediate the excessive inflammatory response and tissue damage [63]. Furthermore, it has been found that in vitro infection with *C. muridarum* rapidly increases CD69 expression on cells and induces the swift production of IFN-γ in NK and NKT cells [64]. Several studies have shown that NKT cells can promote NK cells, and upon stimulation by α-Galcer, NKT cells can enhance the secretion of IFN-γ by NK cells [65].

NKT cells represent an intermediate stage of the immune response, bridging innate and adaptive immunity following exposure to infectious agents. Their rapid response to infection indicates a vital protective function during the early stages of infection, exhibiting diverse functions across various infection models, according to the nature of the infectious factors.

### 2.3. NK Cells

NK cells were initially described as killer effector lymphocytes, which, like other leukocytes, can be recruited to the infected site by chemokines. In contrast to killer T cells, NK cells are capable of directly causing the death of abnormal cells without needing prior sensitization. They release various cytokines, encompassing pro-inflammatory types like TNF-α, as well as immunosuppressive ones such as IL-10. Additionally, NK cells produce a selection of chemokines, including CCL2, CCL3, and CXCL8. These chemokines are essential for enabling the migration of NK cells and various immune cells, such as APCs, to the location of inflammation, facilitating their interactions [66].

NK cell-mediated killing of target cells can affect the T-cell response by promoting the cross-presentation of target cell fragments to CD8^+^ T cells by APCs [67]. Numerous studies have shown that NK cells are considered the major producers of cytokines, especially IFN-γ, in diverse physiological and pathological contexts. In lymph nodes, NK cells produce IFN-γ to help regulate the T-cell response. The cells move from inflammatory tissues in the periphery to lymph nodes that drain these areas, allowing them to engage directly with naive T cells or indirectly through DCs with other immune cells, thus influencing the overall immune response [68].

During the early stage of *Chlamydia* infection, NK cells are considered an important source of IFN-γ. On day 5 post-infection, cytotoxic activity of NK cells is markedly up-regulated in both splenic and pulmonary tissues of severe combined immunodeficiency(SCID) mice, indicating that NK cells are rapidly activated at the onset of pathogen invasion., which lack T and B lymphocytes yet retain functional NK cells, depletion of NK cells with anti-asialo-GM1 antibody results in a significant reduction in pulmonary IFN-γ levels, providing direct evidence that NK cells are an important IFN-γ source [69]. Hook et al. demonstrated that *C. trachomatis* stimulation activates natural killer [70]. During the early stage of genital chlamydial infection, NK cells are not only a primary source of IFN-γ but also pivotal in promoting a protective CD4^+^ Th1 immune response through the secretion of this cytokine [71]. Moreover, this early IFN-γ production by NK cells is also a key mechanism for the clearance of intracellular pathogens like *Leishmania* spp. [72].

However, when NK function is abrogated by anti-asialo-GM1 antibody administered on days 0 and 3 post-infection, no significant differences in pulmonary chlamydial burden (IFU) are observed on days 2, 3, or 5 compared with untreated controls, demonstrating that NK cells do not directly restrict pathogen replication during this early stage [73]. Studies in nude mice demonstrate that early IFN-γ mediated by NK cells can transiently delay infection progression; however, bacterial clearance cannot be accomplished independently in the absence of T cells. Only when Th1 cells are sufficiently expanded and engaged does durable protection ensue [74]. So, these studies suggest that NK cells may not be necessary to resist *Chlamydia* during the innate immune stage. The protective role of *Chlamydia* infection does not stem from the direct control of *Chlamydia* proliferation; rather, it is likely mediated through immune regulation, which enhances the adaptive immune response, thereby assisting in the clearance of *Chlamydia*.

Furthermore, NK cells play a multidimensional role in the immune system that extends far beyond the traditional perception of “direct killers.” The core of their function lies in the precise and bidirectional regulation of adaptive immune responses, depending on the nature of antigenic stimuli. On one hand, NK cells exert inherent cytotoxic effects by directly recognizing and eliminating virus-infected cells and tumor cells. On the other hand, through cell-mediated killing and the secretion of various cytokines (such as IFN-γ and TNF-α), influence innate immune cells including DCs, macrophages, and neutrophils [75]. For instance, NK cells can “edit” DCs to promote their maturation and antigen-presenting capacity, thereby positively regulating the initiation and activation of downstream antigen-specific T and B cell immune responses. Conversely, NK cells can also negatively modulate immune reactions by directly killing overactivated T cells or plasma cells, thus preventing immunopathological damage [75].

Although NK and NKT cells belong to different cell lineages, the two cell types show remarkable similarities. For example, neither can play an effective role without presensitization, but both play a key role in the initial defense against *Chlamydia* infection. However, they play different roles in different types of chlamydial infections, and NKT cells can regulate NK cell activation and cytokine secretion to some extent [64].

### 2.4. The Role of Ex-ILC3s in Chlamydia Infection

ILCs are innate immune cells with adaptive immune functions [76] and play a crucial role within the innate immune system. Although ILCs are found in almost all organs and tissues, they are primarily located in mucosal tissues [77,78,79], where they play a key role in regulating tissue-specific immunity and maintaining immune homeostasis. Although ILCs lack antigen-specific receptors, they can express IL-2Rα (CD25), IL-17Rα (CD127), and other cytokine receptors that receive cytokine stimulation to exert immune function [80]. At present, there is evidence that natural lymphoid cells provide resistance during the early stage of microbial infection to hinder microorganisms from successfully establishing stable colonization within mucosal tissues [81]. *Chlamydia* infection is also regulated by ILCs.

The ILC3s are highly heterogeneous. Acquisition of the transcription factor T-bet in CCR6^−^NKp46^+^ ILC3 initiates a phenotypic switch toward an ILC1-like profile: T-bet represses RORγt activity while directly trans-activating IFN-γ and TNF-α, thereby equipping these cells to combat intracellular pathogens and viruses [82]. Accumulating evidence demonstrates that upon stimulation with cytokines such as IL-12 and IL-2, ILC3s can undergo transformation from cells primarily secreting IL-17 and IL-22 into IFN-γ-producing cells, a process observed both in vitro and in vivo [83]. The identity of ILC3s is maintained by the transcription factor RORγt. Upon stimulation with signals such as IL-12, the expression of the transcription factor T-bet is induced and upregulated, while RORγt is downregulated. Elevated T-bet directly drives IFN-γ production and suppresses the canonical functions of ILC3s. Another critical transcription factor, Aiolos, further reinforces this conversion process. Aiolos is not expressed in resting ILC3s but is highly expressed in intermediate populations during transition and in mature ILC1s [84]. Studies indicate that co-expression of T-bet and Aiolos in vitro synergistically inhibits IL-22 secretion and significantly promotes IFN-γ production, thereby collectively determining the fate transition of ILC3s into ILC1-like cells (also referred to as “ex-ILC3s”). This phenomenon has been directly confirmed in human systems. NKp44^+^ ILC3s isolated from human tonsils downregulate RORγt and upregulate T-bet upon stimulation with IL-12 and IL-15, accompanied by a corresponding shift in their cytokine secretion profile from IL-22 to IFN-γ [85].

Previous data showed that *Chlamydia*, when infected through the reproductive tract, can spread to the gastrointestinal tract and establish long-term colonization [86,87,88], while *C. muridarum* in the reproductive tract is quickly cleared [89,90,91]. Zhong’s study recently identified a mutant *C. muridarum* that lost its ability to persistently colonize the colon of mice [92]. However, this mutant could successfully colonize the colons of mice lacking IFN-γ, prompting the researchers to categorize it as an IFN-γ-susceptible mutant [91]. This evidence shows that IFN-γ plays a key role in *Chlamydia* diffusion, but it cannot determine the specific cellular origin of IFN-γ that inhibits *Chlamydia* infection. Thus, further monitoring in Zhong’s laboratory [93], by transferring this IFN-γ-sensitive mutant strain to IL-7R^−/−^ mice lacking lymphocytes and ILCs, revealed that the colonization ability of *C. muridarum* was restored. However, in Rag1^−/−^ mice lacking lymphocytes but not innate lymphoid cells, the colonization capacity of this strain was not rescued, demonstrating the significant role of IFN-γ derived from ILCs in chlamydial colonization.

Another study [94] has shown that the lack of IL-7R (a common γ receptor-containing receptor required for ILC development) has long since significantly reduced natural immunity to *C. muridarum*. To further identify the relevant ILCs, *Chlamydia* loading was compared in mice with defective transcription factors T-bet or RORγt, as well as in mice lacking both factors following transcervical inoculation. Mice with T-bet deficiency exhibited a significantly higher live yield of *C. trachomatis* in endometrial tissues (including the uterus or uterine horns) on day 3 post-inoculation. Notably, the increase was more pronounced in RORγt^−/−^ mice, suggesting that the ILCs involved are highly dependent on RORγt and likely belong to group 3 ILCs. This suggests that ILC3s play an important role in innate immunity of the endometrium against *C. trachomatis*. At the same time, IFN-γ^−/−^ mice or IFN-γ depletion from Rag1-KO mice significantly reduced innate immunity to *C. trachomatis* infection, suggesting that ex-ILC3s in ILC3s may be crucial for the process of anti-*C. trachomatis.* Because RORγt is a signature transcription factor of ILC3s, ILC3s can be induced to upregulate T-bet and promote IFN-γ production.

ILC3s are an extremely important component of the mucosal immune system. Compared to T cells and B cells, ILCs can rapidly mount immune responses against pathogenic microorganisms invading the intestine and are vital in maintaining intestinal homeostasis. Some researchers have identified ILCs as the primary source of IFN-γ production, noting that only a few IFN-γ-secreting cells are present in Rag2^−/−^ II2rg^−/−^ mice [95]. Recently, ILC3s have also been found to be pivotal in responding to infections by pathogens other than *Chlamydia*. For instance, in colitis caused by *Salmonella* infection, approximately 80% of IFN-γ is produced by Nkp46^+^T-bet^+^ innate lymphocytes, while traditional NK cells account for only 20% of IFN-γ production [96].

### 2.5. T-Cell Resistance Against Chlamydial Infection

Numerous animal studies have demonstrated the vital importance of T cells in clearing chlamydial infections. As early as 1985, Rank et al. reported that T-cell-deficient mice developed chronic infection due to their inability to clear *Chlamydia* post-infection, whereas control wild-type mice successfully cleared infection within 20 days [97]. Conversely, the adoptive transfer of polyclonal *Chlamydia*-specific T cells allowed T-cell-deficient mice to regain the ability to eliminate *Chlamydia* infection [98].

Extensive data suggest that both CD4^+^ and CD8^+^ T cells are involved in controlling chlamydial infection. Both T-cell subsets have been detected at sites of *C. trachomatis* infection in both human and animal models [99]. When *Chlamydia* invades the host, DCs, as the most effective professional APCs, serve as potent antigen precursor cells. They recognize *Chlamydia* antigens primarily through pattern recognition receptors (PRRs), processing pathogen-derived proteins into peptide fragments that form stable antigen peptide-MHC-II molecular complexes [100]. Subsequently, these complexes migrate to secondary lymphoid organs, where they present the antigen peptides to CD4^+^ T cells, thus enabling CD4^+^ T cells to function as sentinels [101]. Furthermore, *Chlamydia* infection is also able to induce specific CD8^+^ T-cell responses. Since the main tissue tropism of *Chlamydia* is epithelial cells, they generally do not express MHC II molecules. Thus, CD8^+^ T cells may play a significant role once the organism infects epithelial cells.

The differentiation fate and subsequent functional commitment of CD4^+^ T helper (Th0) cells are pivotal determinants of infection outcomes in the host immune response against *Chlamydia* [10]. This process is stringently regulated by the integration of pathogen-derived antigens and the host cytokine milieu. APCs process and present major outer membrane protein (MOMP) and other chlamydial antigens, driving the polarization of Th0 cells toward a Th1 phenotype under the influence of key cytokines such as IL-1 [102]. Activated Th1 cells predominantly secrete IL-2, IFN-γ, and TNF-α. Among these, IFN-γ serves as the central effector cytokine responsible for suppressing intracellular chlamydial replication. It mediates antimicrobial effects primarily by inducing host cell expression of indoleamine 2,3-dioxygenase (IDO), leading to tryptophan starvation, and stimulating nitric oxide (NO) production [103]. Furthermore, the Th1 immune response promotes immune polarization through IFN-γ–mediated suppression of Th2 cell proliferation and function [104]. Conversely, Th0 cells can differentiate into Th2 cells under the influence of IL-4, chlamydial outer membrane protein 2 (Omp2), and CHSP60 [105]. Activated Th2 cells secrete cytokines such as IL-4, IL-5, and IL-10, which potently stimulate humoral immunity. These cytokines facilitate B cell proliferation, antibody class switching, and the production of significant levels of immunoglobulins, including IgG and IgA [102]. These antibodies contribute to immune protection primarily at mucosal surfaces by neutralizing extracellular elementary bodies (EBs), thereby preventing their attachment to and invasion of host cells [98].

Cytokines secreted by Th1 and Th2 cells play an important role in anti-*Chlamydia* infection [103]. These cytokines not only regulate the differentiation of Th0 cells into Th1 or Th2 cells but also play their respective immune functions. During anti-*Chlamydia* infection, cytokines play a pivotal role in modulating the host’s defense mechanisms. Animal experiments have confirmed that Th1 cell-mediated cellular immunity plays a central role during host cell clearance of *Chlamydia* infection, while Th2 cell-mediated humoral immunity serves a supportive role. Th1 cells provide activation signals to macrophages by expressing membrane molecules such as CD40L and secreting cytokines like IFN-γ, while activated macrophages can further enhance the effect of Th1 cells by upregulating immune molecules like CD80, CD86, and MHC, as well as secreting additional cytokines, including IL-12 and IL-2, alongside IFN-γ.

Numerous studies have shown that the IFN-γ produced by Th1 cells is essential for defense against *Chlamydia* infection. Moreover, the key mechanism by which CD8^+^ T cells control *Chlamydia* proliferation is through the secretion of IFN-γ. In addition to its antimicrobial mechanisms, IFN-γ secreted by innate immune cells activates the antibacterial effects of mononuclear phagocytes and promotes T-cell responses, thus controlling bacterial dissemination in mucosal tissues. Chlamydial antigens that evade clearance by the innate immune system are captured and processed by APCs, such as dendritic cells. Upon maturation and migration to the draining lymph nodes, these cells present processed peptides to naïve T lymphocytes via MHC, thereby initiating the adaptive immune response [106]. This process triggers the clonal expansion and differentiation of antigen-specific T cells, leading to the generation of effector Th1 cells and cytotoxic CD8^+^ T lymphocytes. These antigen-experienced T cells then traffic to the site of infection, where they mediate pathogen clearance primarily through the secretion of IFN-γ and cell-mediated cytotoxicity. After the antibody blocks the function of CD8^+^ T cells and CD4^+^ T cells, the symptoms of *Chlamydia* infection are significantly aggravated, affecting the prognosis, and IFN-γ is the main effector.

## 3. How Does the Induced IFN-γ Affect *Chlamydia* Infection and Pathogenesis

The biological activity of IFN-γ is very broad and plays a crucial role in immunomodulatory processes and immune response. Comprehensive studies in the *C. muridarum* murine model consistently establish cell-mediated immunity as the predominant host defense mechanism against chlamydial infection. Following infection, immune cells are activated and secrete abundant IFN-γ, effecting pathogen clearance [107]. Leonhardt et al. further demonstrated, in a *C. muridarum* genital tract infection model, that IFN-γ not only prevents *Chlamydia* from entering a persistent state but also markedly diminishes the risk of reactivation [108]. Collectively, these findings define IFN-γ as the critical effector cytokine for eradication of chlamydial infection.

Many in vitro experiments have proven that high concentrations of IFN-γ can completely inhibit the growth of *Chlamydia*, while medium and low concentrations disrupt normal growth [109,110]. In a systematic in-vitro study, Beatty et al. rigorously demonstrated that IFN-γ exerts a dose-dependent inhibitory effect on the growth of *C. trachomatis*. At concentrations ≥ 100 U mL^−1^, no typical inclusions were detected after 48 h of culture, indicating complete growth suppression. At intermediate-to-low IFN-γ concentrations (0.3–30 U mL^−1^), the organism remains viable but exhibits a 50–90% reduction in inclusion number and undergoes rapid morphological conversion into enlarged, aberrant reticulate bodies (RBs). These atypical RBs are non-infectious yet metabolically active, as evidenced by up-regulated expression of the 60-kDa CHSP60 and marked down-regulation of the MOMP. Subsequent experiments revealed that, upon IFN-γ withdrawal and restoration of adequate tryptophan levels, these aberrant RBs resume normal binary fission within 24–48 h and ultimately differentiate into a large population of infectious EBs, confirming their reversible persistent phenotype [111].

### 3.1. Mechanism of Action of the IFN-γ

IFN-γ exerts its anti-*Chlamydia* activity through multiple effector pathways; however, the predominant operative mechanisms differ fundamentally between human and murine systems. In human cellular systems (encompassing epithelial, macrophage, and fibroblast lineages), the central mechanism is the IFN-γ-inducible IDO pathway [111]. In contrast, within murine macrophage models, the IDO pathway is non-essential for protective immunity. IFN-γ-dependent immunity is primarily driven by the upregulation of inducible nitric oxide synthase (iNOS), which catalyzes the production of high levels of nitric oxide (NO), conferring direct cytotoxicity against chlamydial inclusions [112]. Furthermore, IFN-γ modulates the expression of iron homeostasis genes (e.g., downregulation of the transferrin receptor, TfR), inducing an intracellular iron restriction state that starves the pathogen of iron, an element crucial for its replication [113] (Figure 2). The iNOS and iron-withholding pathways act synergistically to form key effector mechanisms for pathogen clearance in murine models. In conclusion, IDO-mediated tryptophan catabolism constitutes the cornerstone of IFN-γ-dependent anti-chlamydial immunity in humans, whereas in murine models, this role is served collectively by the iNOS/NO pathway and iron sequestration mechanisms. This remarkable species-specific divergence stems from differential regulation of the IDO and iNOS gene promoters in response to IFN-γ signaling [114]. Acknowledging this dichotomy is critical for the accurate interpretation of preclinical data from animal models and their subsequent translation into human pathophysiological insights.

#### 3.1.1. IDO Pathway

IDO is the rate-limiting enzyme that catalyzes the decomposition of tryptophan into formyl kynurenine and kynurenine [115,116]. After *Chlamydia* infects human cells, IFN-γ exerts its anti-chlamydial effects primarily through the induction of IDO, the rate-limiting enzyme that catalyzes the degradation of tryptophan into kynurenine, thereby depleting the host cell’s essential tryptophan pool [115,116]. This state of “tryptophan starvation” selectively targets and potently inhibits the replication of obligate intracellular pathogens that are strictly auxotrophic for tryptophan, including *C. trachomatis* (non-LGV serovars) and *C. pneumoniae* [111]. It is critical to note that this efficacy is contingent upon the evolutionary loss of tryptophan biosynthesis operons in these strains—a genomic reduction resulting from reductive evolution during long-term adaptation to the nutrient-rich intracellular niche, rather than a direct physiological response to IFN-γ [2]. The metabolic constraint imposed by IDO activation disrupts the chlamydial developmental cycle, arrreticulateesting bacterial replication and driving the formation of aberrant, non-infectious bodies (aberrant bodies) that may persist intracellularly in a viable but non-cultivable state. This bacteriostatic effect is reversible in vitro; exogenous tryptophan replenishment restores productive bacterial growth, underscoring the metabolite-dependent nature of this host-pathogen interaction [109]. Furthermore, IDO expression is synergistically enhanced by pro-inflammatory signals such as LPS, TNF-α, and IL-1 in concert with IFN-γ [117], highlighting the integration of this mechanism within broader inflammatory and immune networks.

IL-1β inhibits *Chlamydia* replication by upregulating IFN-γ-induced IDO activity, thereby depleting tryptophan. On the one hand, it promotes the synthesis of interferon regulatory factor-1. On the other hand, it upregulates the expression of IFN-γ receptor (IFN-γR) through the NF-κB (p65/p50 heterodimer) pathway. Nuclear NF-*κ*B can directly bind to the *κ*B site of the IFN-γR promoter region to enhance its transcription. Increased sensitivity of cells to IFN-γ (seen in HeLa cells, islet β cells, etc.) [118,119]. However, some cells may express “decoy receptors” to bind IFN-γ but do not activate the signal transducer and activator of transcription-1 (STAT-1)-IDO pathway, thereby reducing the local effective concentration of IFN-γ and inhibiting tryptophan depletion, thereby facilitating immune evasion by *Chlamydia* [119]. Notably, although IL-1β-deficient mice showed high vaginal bacterial load and delayed clearance, the pathological damage of fallopian tubes was significantly reduced (the incidence of hydrops was reduced from 70% to 30%), suggesting that IL-1β is a key factor in controlling the process of infection and determining the pathological outcome [120].

#### 3.1.2. IFN-γ-Induced Activation of the iNOS Pathway

IFN-γ can induce mouse epithelial cells, macrophages, and fibroblasts to secrete iNOS, which hydrolyzes L-arginine to produce NO, thereby irreversibly inhibiting the growth of *Chlamydia*. iNOS catalyzes the production of various reactive nitrogen intermediates, particularly NO [121,122]. NO is an important defense molecule against bacterial pathogens and inhibits the growth of *Chlamydia* in vitro. However, genetic evidence reveals that iNOS-deficient (iNOS^−/−^) mice exhibit significantly greater resistance to *Chlamydia* than IFN-γ receptor-deficient (IFN-γR^−/−^) mice. This phenotypic disparity unequivocally indicates that IFN-γ activates additional, antimicrobial mechanisms beyond the iNOS/NO pathway.

#### 3.1.3. IFN-γ-Induced Iron Deficiency Pathway

Iron serves as an essential micronutrient for both mammalian hosts and invading bacterial pathogens, being integral to numerous metabolic processes. To counteract infections, hosts employ a defense mechanism known as nutritional immunity, strategically limiting the availability of iron to impede bacterial growth [123,124]. IFN-γ has an important effect on iron metabolism. Studies have shown that IFN-γ restricts iron availability througha a variety of mechanisms: it downregulates the expression of transferrin receptor (TFR) in infected host cells to limit cellular uptake, downregulates the expression of mononuclear macrophage ferroportin (FPN) to inhibit iron release from macrophages, thereby reducing circulating iron, and upregulates the expression of divalent metal ion transporter 1 (DMT1) in monocyte macrophages to enhance iron absorption, further reducing circulating iron [125,126,127].

## 4. Discussion

Upon infection with *Chlamydia*, the activation state of epithelial cells is altered, and they produce various chemokines and inflammatory cytokines. These factors drive the inflammatory response and promote the recruitment of various immune cells to the infection site. Subsequently, both innate and adaptive immune responses are activated, forming a key line of defense against microbial infections. As a classic and representative inflammatory factor, IFN-γ plays a crucial role in all stages of host control during chlamydial infection.

In this review, we speculate that the IFN-γ released by different cells may play different roles at different stages of *Chlamydia* infection and in different types of *Chlamydia* infection processes (Figure 3). First, the different stages of chlamydial infection correspond to the sequential activation of immune cells. In the early stage of infection, NK cells and NKT cells, as the key components of innate immunity, can rapidly secrete IFN-γ and play an initial defense role in preventing the systemic dissemination of *Chlamydia*. NKT cells can also regulate the phenotype and function of DC and initiate the subsequent CD4^+^ and CD8^+^ effector T cell responses. Cytokines such as IL-12 and IL-15 in local tissues can induce some ILC3s to down-regulate RORγt and up-regulate T-bet, transforming them into IFN-γ-secreting ex-ILC3s, which are important cell sources for inhibiting *Chlamydia* colonization in the large intestine and endometrium. In the adaptive immune stage, IFN-γ secreted by Th1 (CD4^+^) cells is the core effector against *Chlamydia* infection, while CD8^+^ T cells become the key force to control the proliferation of *Chlamydia* by secreting IFN-γ. Both of them participate in pathogen clearance in the late stage of infection. Second, the chlamydial species and the route of infection determine the type of immune cell that plays a role: In a mouse lung infection model, *C. pneumoniae* infection is more likely to induce NKT cells to produce IFN-γ, while *C. muridarum* infection is more likely to induce NKT cells to secrete IL-4, which directly affects the differentiation direction of downstream T cell subsets. In a model of genital tract infection, IFN-γ produced by CD4^+^ T cells was shown to be critical for controlling infection and preventing tubal disease. In contrast, ex-ILC3s-derived IFN-γ specifically inhibited *Chlamydia* colonization in large intestine and endometrial infections.

Meanwhile, in addition to exerting antibacterial immunity through the IDO pathway, activating the iNOS pathway, and inducing the iron deficiency pathway, IFN-γ may also play a significant role in reshaping the signaling network of host cells. Recent studies have revealed that *Chlamydia* can affect the host Akt-mTORC1 axis: it maintains mTORC1 activity by phosphorylating TSC2 or PRAS40, thereby inhibiting autophagy and evading degradation by being encapsulated in the autophagosome-lysosome pathway [128,129]. Therefore, when the host is in a state of high expression of IFN-γ, the *Chlamydia*-dependent Akt-mTORC1 immune escape mechanism is inhibited, forcing the *Chlamydia* to enter the autophagy-lysosomal degradation pathway and eventually be cleared by the host cells.

Notably, in the murine system, the IL-22 gene resides on chromosome 10, close to IFN-γ. It has been shown that transcription activators, such as Gcn4, regulate the expression of neighboring genes [130]. Therefore, changes in IL-22 expression may affect the transcriptional level of IFN-γ, but at present this mechanism has not been experimentally verified and needs to be further investigated. At the same time, the data [131] show that in the process of *Chlamydia* respiratory infection, IL-22 is the key cytokine that determines the growth and proliferation of various T-cell subsets, such as Th17 cells, especially early in the infection, which may also involve γ*δ*T cells, NKT cells, and others. Excessive IFN-γ can cause tissue damage, and the host can activate the negative regulatory axis with IL-37 as the core to achieve a precise “brake” by inhibiting the secretion of IL-12/IL-18 in dendritic cells, blocking the activation of NLRP3 inflammasome, and promoting the polarization of macrophages to M2 type, while retaining the antibacterial effect of IFN-γ [132,133,134,135]. It can effectively reduce the tissue damage caused by it. This “attenuation and synergism” regulatory mode provides a new immune intervention strategy for *Chlamydia* infection.

Although the role of IFN-γ in different stages of chlamydial infection has been extensively studied, there are still some important questions that require further exploration. For example, how does IFN-γ secreted by different cells coordinate their role in chlamydial pathogenesis? *Chlamydia* is absent from the small intestine for approximately two weeks but it can persist in the large intestine for several months. What role do different microbial flora play in this process? ILC1s are also an important part of mucosal immunity and can secrete IFN-γ. However, in the context of *Chlamydia* infection of the colon and uterus, IFN-γ released by ILC3s appears to play a major role. Is this difference attributable to distinct mechanisms of delivery to the site of infection?

At the same time, IL-22 acts as an immune mediator produced by specific cell types and targets particular tissue cells. Its effects on different tissue cells appear to be largely similar (enhanced innate immunity, damage protection, and improved regeneration), though the nature of its impact (both protective and pathogenic) is influenced by the tissue’s location, the condition of the affected area, and the surrounding cytokine milieu. Additionally, since IL-22 may boost the effects of IL-1β, TNF-α, IL-17, and IFN-γ, exploring the potential of combining IL-22 with IFN-γ for the purpose of developing *Chlamydia* vaccines would be a valuable area of research.

## Figures and Tables

**Figure 1 microorganisms-13-02374-f001:**
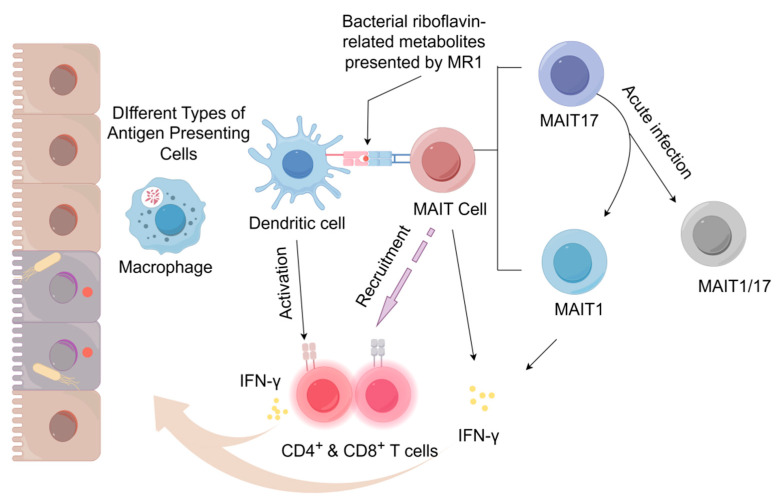
MAIT Cell Activation and IFN-γ Production in Response to Bacterial Infection. During infection, MAIT cells are activated via the interaction of TCR with riboflavin metabolites presented by MR1 molecules on professional APCs. Simultaneously, pro-inflammatory cytokines released by APCs synergistically stimulate T cells. Activated MAIT cells promptly secrete IFN-γ and recruit CD4^+^ and CD8^+^ T cells to the site of infection. Notably, MAIT cell subsets demonstrate functional plasticity; for instance, MAIT17 cells can transition into MAIT1 or MAIT1/17 phenotypes under acute infection conditions, thereby acquiring the capacity to produce IFN-γ, which is essential for eliminating intracellular pathogens. Created with https://www.figdraw.com (accessed on 20 June 2025).

**Figure 2 microorganisms-13-02374-f002:**
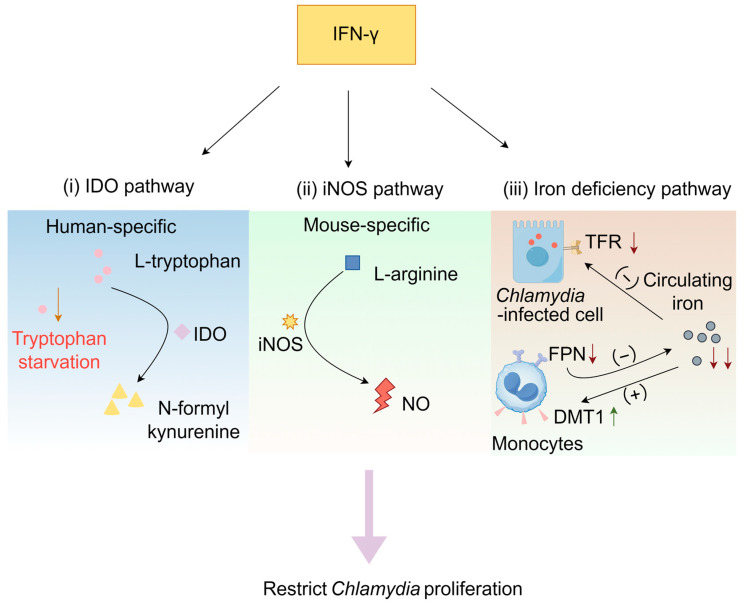
IFN-γ inhibits *Chlamydia* growth through the IDO pathway, iNOS pathway, and iron deficiency pathway. (**i**) After *Chlamydia* infects human cells, IFN-γ can induce the activation of IDO, resulting in the conversion of L-tryptophan to N-formylkynourine. This process depletes tryptophan in the cellular microenvironment, leading to a “tryptophan starvation” state. (**ii**) IFN-γ can induce mouse cells to secrete iNOS. iNOS hydrolyzes L-arginine to generate NO. (**iii**) IFN-γ down-regulates the expression of TFR in infected cells and inhibits the uptake of iron by cells; IFN-γ down-regulates the expression of FPN in mononuclear macrophages, inhibits the release of iron by monocytes, and reduces the source of circulating iron; IFN-γ upregulates the expression of DMT1 in mononuclear macrophages, promotes the uptake of iron by mononuclear macrophages, and further reduces circulating iron. Created with https://www.figdraw.com (accessed on 21 June 2025).

**Figure 3 microorganisms-13-02374-f003:**
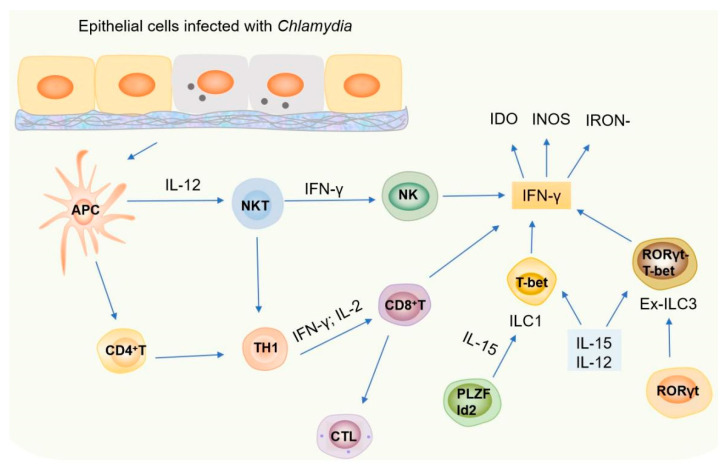
Immune responses to *Chlamydia* infection. *Chlamydia* primarily infects epithelial cells, triggering the production of pro-inflammatory cytokines (e.g., IL-12, IL-15) and chemokines. These mediators recruit and activate innate and adaptive immune cells, including NK cells, T cells, NKT cells, and ILC subsets (ILC1/ILC3), which drive IFN-γ production. IFN-γ restricts bacterial growth via IDO, iron deprivation, and iNOS pathways.

## Data Availability

No new data were created or analyzed in this study. Data sharing is not applicable to this article.

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
