# Peer review of "Role of IFN-γ from Different Immune Cells in *Chlamydia* Infection"

_microorganisms, 2025, doi:10.3390/microorganisms13102374_

Round 1

Reviewer 1 Report

Comments and Suggestions for Authors

Title: Role of IFN-γ from different immune cells in Chlamydia Infection.

In this paper the authors study innate and adaptive immunity against chlamydial infection. The authors focus their attention on gamma interferon (IFN-γ), a defence cytokine against chlamydial infection. The authors state that these immune mechanisms against chlamydia are poorly understood.

This paper seems good to me; however, I have some concerns.

The abstract is missing the conclusion.

Studies indicate that IL-1β stimulates the transcription of the IFN-γR receptor via activation of the NF-κB pathway, thereby increasing the sensitivity of cells to IFN-γ. IL-1β is crucial for clearing the organism of Chlamydia: animal models (C. muridarum and C. trachomatis) show that the absence of IL-1β prevents effective eradication of the bacterium and improves pathogen clearance. However, excessive activation of IL-1 may contribute to tissue damage. IFN-γ, produced by Th1 cells, inhibits intracellular replication by inducing enzymes such as indoleamine 2,3-dioxygenase (IDO). IL‑37: serves as a brake, limiting inflammation and guiding resolution, with significant therapeutic promise.

Toniato E. IL-37 is an inhibitory cytokine that could be useful for treating infections . International Journal of Infection. 2024;8(1):1-2. (www.biolife-publisher.it).

Akt typically activates mTORC1 by phosphorylating TSC2 or PRAS40, promoting growth and inhibiting autophagy. Chlamydia decreases host autophagic flux, potentially via an active mTOR pathway, which prevents it from being sequestered in autolysosomes. In the light of these concepts, to make this paper more interesting for the readers of this important journal, the authors should expand a bit the discussion (or introduction). Below I report an interesting article that should be studied, incorporate the meaning and report it briefly in the discussion and in the list of references.

Avivar-Valderas A. Inhibition of PI3Kβ and mTOR influence the immune response and the defense mechanism against pathogens. International Journal of Infection. 2023;7(2):46-49. (www.biolife-publisher.it).

Figure 3 is too complex (many cells and many cytokines). It is recommended to divide it into 2 parts.

I believe these suggestions are important for improving this paper. Without these corrections the paper cannot be published. So, I recommend minor revision.

Comments on the Quality of English Language

The English could be improved

Reviewer 2 Report

Comments and Suggestions for Authors

The manuscript “Role of IFN-γ from Different Immune Cells in Chlamydia Infection” by Yang et al. proposes a theory about the participation of IFN-γ in the pathogenesis of disease caused by this bacterium, which could depend on the cell type. IFN-γ is capable of producing interferon gamma (IFN-γ) as a defense mechanism against chlamydial infection, thereby effectively mediating the clearance of infection. Under the conditions presented, the article is highly speculative since it does not provide experimental evidence or clinical data suggesting that IFN could participate in the evasion/permissiveness of infection and/or symptoms. The specific question posed in discussion L539 “IFN-γ secreted by different cells coordinate their role in chlamydial pathogenesis?” is not answered in this manuscript.

L17 states “However, the distinct contributions of various immune cell populations in response to chlamydial infection as well as the specific functions of these cell types at different stages of infection remain poorly understood” perhaps suggesting that there are structural variations in IFN-γ expression. “various roles of IFN-γ released by different immune cells in chlamydial infection” could be more specific? The role of IFN-γ receptors is not considered relevant in the context of the work and is not mentioned.

Some ideas are not adequately stated, for example, L46-47, “cytokines IL-1, IL-6, TNF-α, and IFN-γ activate or attract immune cells to initiate or enhance inflammation in response to Chlamydia.” It would be important to identify whether these ILs possess chemoattractant activities.

The section related to MR1 and its potential interaction with metabolites produced by the bacteria. Interestingly, MR1 is a monomorphic major histocompatibility complex class I-related molecule that is markedly conserved in diverse mammalian species. MAIT cells are not present in germ-free mice, indicating that commensal flora is required for their expansion in the gut lamina propria. MR1-restricted MAIT cell selection is dependent upon B cells, and their accumulation in the gut lamina propria and mesenteric lymph node requires the commensal bacterial flora. Therefore, they are thought to play an essential role within the immune system, in which IFN-γ could actively participate.

A diagram related to the potential cellular activators that promote the synthesis and expression of IFN-γ would be interesting to include in an attempt to understand why there might be some potential difference in IFN-γ synthesis.

There are also highly speculative approaches that require further analysis and experimental bases. For example, in L 530-531, it is indicated that there is a relationship due to the location of the IL-22 and IFN-γ genes on chromosome 10; thus, changes in IL-22 expression may affect the regulation of IFN-γ gene expression, but this is only a hypothesis.

The potential role of endogenous components of Chlamydia as potential activators of IFN synthesis is not specified.

Reviewer 3 Report

Comments and Suggestions for Authors

Role of IFN-γ from Different Immune Cells in Chlamydia Infection

I congratulate the authors on a well-done review of the activation and function of Interferon-gamma in the development of C. trachomatis during infection. I want to comment on two points. Although Chlamydia trachomatis possesses its own enzymatic systems, it not only requires ATP, but many vital nutrients are synthesized solely by the host, which has complex implications. This intricate relationship means that the bacterium not only needs the triphosphate and intermediate metabolites from the host cell as energy sources for growth, but also a range of metabolites, such as tryptophan, phospholipids, and nucleotides, for its survival. This highlights that Chlamydia trachomatis has evolved to minimize its genome size at the expense of complete independence, making it highly dependent on its host.

The second point is that there is a significant gap in our understanding regarding the role of MAIT cells in Chlamydia infection or the riboflavin metabolism of this bacterium. This is an area that urgently requires further investigation. In this section, the authors do not suggest any mechanism for involvement. Have you considered that riboflavin acts as a precursor to the coenzymes FMN and FAD, which participate in redox reactions, such as electron transport and cellular respiration?

Chlamydia are also auxotrophic for biotin, lipoic, and pantothenic acids. Chlamydia may subvert the mammalian sodium multivitamin transporter (SMVT) to move multiple essential compounds into the inclusion, where BioY (CTL0613) and other transporters would be present to facilitate transport into the bacterium.  Riboflavin may have a similar situation, where the riboflavin vitamin transport 3 (RVT3) is modified for riboflavin transportation to the inclusion vesicle. So, the distinctive capability of MAIT cells to recognize metabolites derived from the vitamin B biosynthesis pathways in bacteria would not activate MAIT cells. (Fisher DJ, Fernández RE, Adams NE, Maurelli AT. Uptake of biotin by Chlamydia Spp through the use of a bacterial transporter (BioY) and a host-cell transporter (SMVT). PLoS One. 2012;7(9):e46052. doi:10.1371/journal.pone.0046052).

What do you think about it?

 Intestinal macrophages are susceptible to Chlamydia infection, as evidenced by the presence of Chlamydia in enteroendocrine cells and macrophages of the small bowel in patients with severe irritable bowel syndrome. MAIT cells may play a role in responding to Chlamydia infection, but this has not been proven. (Dlugosz A, Törnblom H, Mohammadian G, Morgan G, Veress B, Edvinsson B, Sandström G, Lindberg G. Chlamydia trachomatis antigens in enteroendocrine cells and macrophages of the small bowel in patients with severe irritable bowel syndrome. BMC Gastroenterol. 2010;10:19. doi: 10.1186/1471-230X-10-19).

Round 2

Reviewer 2 Report

Comments and Suggestions for Authors

The masnuscript is better improved now and could accepted